# COVID-19 Vaccination: From Interesting Agent to the Patient

**DOI:** 10.3390/vaccines9020120

**Published:** 2021-02-03

**Authors:** Anis Daou

**Affiliations:** 1Pharmaceutical Sciences Department, College of Pharmacy, QU Health, Qatar University, Doha P.O. Box 2713, Qatar; adaou@qu.edu.qa; 2Biomedical and Pharmaceutical Research Unit, QU Health, Qatar University, Doha P.O. Box 2713, Qatar

**Keywords:** COVID-19, vaccination, severe acute respiratory syndrome coronavirus 2 (SARS-CoV-2), drug development and discovery, pandemic

## Abstract

The vaccination for the novel Coronavirus (COVID-19) is undergoing its final stages of analysis and testing. It is an impressive feat under the circumstances that we are on the verge of a potential breakthrough vaccination. This will help reduce the stress for millions of people around the globe, helping to restore worldwide normalcy. In this review, the analysis looks into how the new branch of Severe Acute Respiratory Syndrome Coronavirus 2 (SARS-CoV-2) came into the forefront of the world like a pandemic. This review will break down the details of what COVID-19 is, the viral family it belongs to and its background of how this family of viruses alters bodily functions by attacking vital human respiratory organs, the circulatory system, the central nervous system and the gastrointestinal tract. This review also looks at the process a new drug analogue undergoes, from (i) being a promising lead compound to (ii) being released into the market, from the drug development and discovery stage right through to FDA approval and aftermarket research. This review also addresses viable reasoning as to why the SARS-CoV-2 vaccine may have taken much less time than normal in order for it to be released for use.

## 1. Introduction

### 1.1. Covid-19 Breakdown and Background

Severe Acute Respiratory Syndrome Coronavirus 2 (SARS-CoV-2)—better known as Coronavirus or COVID-19—was first encountered in the capital city of the Hubei province of Wuhan, China. In late December 2019, health authorities began to identify unknown viral pneumonia cases that started to spread to other parts of China. Due to how much it had spread by early January 2020, an identification technique known as Reverse Transcription Polymerase Chain Reaction (RT-PCR) was applied, which enabled scientists in real-time to establish a diagnosis for these cases by way of distinguishing and isolating the novel Coronavirus from which viral pneumonia the patients were suffering was caused [1].

By the end of January 2020, after succumbing to considerable pressure the World Health Organization (WHO) officially declared COVID-19 as a pandemic, a Public Health Emergency of International Concern (PHEIC). Due to action that was not taken sooner in many parts of the world, the virus spread to around 25 countries by early February 2020. With the numbers increasing, guidelines and criteria for diagnosis, treatment and preventative measures had to be established rapidly [1,2]. Viral detection using RT-PCR identified the SARS-CoV-2 virus to be the disease which caused this viral transmission worldwide. This virus bore significant similarity to that found present within bats and was of the same family as Severe Acute Respiratory Syndrome Coronavirus 1 (SARS-CoV-1) and Middle East Respiratory Syndrome Coronavirus (MERS-CoV), therefore significantly narrowing down the likelihood that this had been somehow transmitted from bats to humans (owing to bats being the main reservoir for this virus). COVID-19 has a high recombinant and mutation rate due to its unique replication capabilities, enabling it to adapt to new host cells and different target sites [3]. So far, Covid-19 has been defined by 17 known mutations (14 non-synonymous mutations and 3 deletions), eight of these mutations have been on the spike protein, the main target site for the vaccination, with at least three of these mutations having a significant biological effect. These mutations, in particular N501Y, can incur a substantial change in the binding domain, resulting in enhancing the binding affinity to the human ACE2 enzyme. Another mutation (P681H) that is located directly close to the spike protein has shown the potential to increase infection and transmission. In terms of deletions that have occurred to the viral genome, the deletion of two amino acids has shown a link indicating immune escapability in immunocompromised patients thus enhancing viral infectivity [4]. The transmission of COVID-19 is from human-to-human contact [5], the most common infections occur from sufferers who are asymptomatic, therefore transmitting the virus without being aware they are carriers [6].

Symptoms of COVID-19 as was mentioned consist of two states, the (i) symptomatic state and the (ii) asymptomatic state. The symptomatic state can be easily noticed through the patient showing multiple different symptoms, one of them being the Acute Respiratory Disease Syndrome (ARDS), which include fever, cough, tiredness, sore throat, headache, and myalgia. More severe symptoms include aches and pains, diarrhoea, conjunctivitis, loss of taste and smell, a rash on skin and discoloration of fingers or toes; the most severe cases include difficulty breathing, chest pains or pressure and can even lead to loss of speech and movement [7]—some of the symptoms can result in multiple organ failure and eventually death. ARDS patients who experience symptoms tend to carry underlying health conditions, a suppressed immune system or are of older age. According to the literature, asymptomatic COVID-19 sufferers are the main source of transmission; through their respiratory droplets being airborne, as well as transmitted through virus-contaminated containers and foods [8,9]. Asymptomatic carriers show no symptoms of the virus due to an immune system capable of combatting the virus. However, they are capable of infecting others, henceforth making the virus capable of spreading around and becoming sometimes untraceable. The only way to identify an asymptomatic patient is through the administration of an RT-PCR. This, therefore, makes it difficult for countries to conduct identification tests whilst attempting to control the spread of the virus [10].

The structure of SARS-CoV-2 has shown to have a single strand enveloped RNA (sRNA). The size of the virus is between 50 and 150 nm in diameter, its linearity and positive-sense RNA genome is large. It belongs to the CoV-2 family, which was firstly found in the mid 1960s [11,12]. This family of viruses has a spherical shape, with envelopes containing helical nucleo-capsids and nucleoproteins; these are associated with the genomic structure of RNAs. The virus is capable of attaching itself to the host cells of its target due to a trimer of spike glycol-proteins, which include hemagglutinin esterase; there are also integral membrane and envelope proteins [13]. The virus can infect humans. Although animals will have varying immune responses to the virus, their immune system is more equipped to combat it and therefore it does not spread between animals as quickly as it does between humans. The virus targets the respiratory, hepatic, gastro-intestinal and neurological systems [12]. The name Coronavirus is due to the presence of the crown-like structures identified when scanned under the electron microscope. The viral structure consists of envelopes that contain helical nucleo-capsids and nucleoproteins (N), these are associated with the RNA genome. Embedded in the envelope is a 2 nm trimer of spike glycoproteins (S), this is the main source of the virus’s attachment to the receptor of the host cells. Within the virus, it also consists of integral membranes (M) and envelope proteins (E). Beta-Coronaviruses have an additional membrane glycoprotein named hemagglutinin esterase, which contains 5–7 nm long spikes (See Figure 1). There are different families of the Coronavirus; the International Committee on Taxonomy of Viruses (ICTV) has separated them into different genres depending on their activity and structure. The genres are named; Alpha, Beta, Gamma and Delta Coronaviruses [14]. Several human Coronaviruses (alpha-CoVs, HCoVs-NL63, beta-CoVs, HCoVs-OC43, HCoVs-229E, HCoVs-HKU1, MERS-CoV, SARS-CoV and ARDS have been identified [15]. New versions of the Coronavirus will appear due to the large genomic potential, rapid mutation capabilities, high prevalence and wide distribution within the bird and animal kingdom. The emergence of CoVs is due to birds being able to carry this viral form and transfer it from area to area through flying and being capable of inhabiting in between groups [16,17].

SARS-CoV-2 binds to the host cell Angiotensin-Converting Enzyme 2 (ACE 2) through its spike proteins. These spike proteins consist of two subunits, the receptor-binding subunit, which facilitates binding of the virus to the host cell, and the membrane fusing subunit, which allows for the fusing of the membranes of the virus and host cell. Once the virus binds to the host cell, the viral molecule then enters into the host cell [19]. Prior to entering the cells, the protease enzyme TMPRSS2 activates the spike proteins. The combination of activation and binding to ACE2 are required for successful admission into the cell [20]. Once entered they translate small parts of the virus onto non-structural proteins. The proteins then form an enzyme called RNA Dependent RNA Polymerase. The enzyme induces a double membrane vesicle, through the restructuring of the endoplasmic reticulum of the cell. Once these vesicles are formed, continuous replication and transcription are made of the sRNA gene coding for SARS-CoV-2 [21]. Once this is complete all the viral proteins with the sRNA are collated in the endoplasmic reticulum and the Golgi apparatus of the cell, hence forming the new virus particle. These viruses then release and spread within the body, attacking target sites as mentioned above [22].

The most common test to detect the virus is through RT-PCR. This test requires both a nasal and throat swab. The test detects the RNA of the virus, which may be present within a patient prior to the formation of antibodies or symptoms. With this test, early-stage detection can be achieved. The RT-PCR targets two parts to the virus, the Open Reading Frame Gene (ORFG) and the viral nucleo-capsid regions. The test works through the reverse transcription of the RNA of the virus into a complementary DNA (cDNA) [21]. This is then amplified in the Real-Time Polymerase Chain Reaction thermal cycle. The dye used produces fluorescent signals, whereby the RT-PCR is then capable of automatically forming a curve, thus giving a quantitative analysis for the presence of the SARS-CoV-2 virus at the nucleic acid level. RT-PCR can detect the virus in asymptomatic persons, however, the test is capable of giving false-negatives hence patients may be tested twice before being confirmed as positive or negative [21,23]. The RT-PCR kit remains an effective kit to use in the identification of SAR-CoV-2, however, one of the main worries for challenges arising from this analysis is that cases may have gone undetected; several studies have shown that the clinical sensitivity of the analysis of respiratory swabs was at around 70% effective. This was due to the timing of these swabs, the type of specimen obtained and the quality of the sample taken. The viruses present in the upper respiratory tract for the first several days following the onset of symptoms, hence after 5 days of symptoms, it becomes increasingly difficult to identify the virus via RT-PCR. In the latter stages, for correct and accurate reading, swabs from the lower respiratory tract will yield a higher rate of detection. Due to these nuances, it has been challenging laboratory professionals to truly define the clinical sensitivity of SARS-CoV-2 real-time PCR and has required that negative results be interpreted in the context of the timing of the sample [24]. An example of problems faced for SAR-CoV-2 RT-PCR testing is shown in Figure 2.

### 1.2. The Research and History Ortho-Coronavirinae

Coronaviruses constitute the subfamily *Ortho-Coronavirinae.* In this section, we look at the history of this viral family, the research that has been undertaken and what we know about the coronaviruses. Figure 3 shows the history of the coronaviruses.

### 1.3. Vaccinations Proposed for SARS-CoV-1

Upon the outbreak of SARS-CoV-1, several research projects were launched, this led to multiple potential vaccines showing promise, these included the following:Inactivated SARS-CoV based vaccine: this vaccine expressed several structural proteins such as nucleo-capsid, membranes and spike proteins [27]. These are thought to induce an immune reaction that is capable of stimulating an immune response. The inactivated virus was intended for use as a first-generation vaccine, this is due to the ease of generation of these inactivated viral particles. The next step was the replacement of the inactivated viral vaccine by a second vaccine based around fragments containing neutralizing epitopes that are safer and more efficacious to use. Several reports have shown that SARS-CoV-1 was inactivated with formaldehyde, UV light, and β-propiolactone which can induce virus-neutralizing antibodies in immunized animals [28,29,30].S-protein based vaccines: several recombinant based vaccines that have expressed the spike protein in SARS-CoV-1 were assessed in pre-clinical studies [31]. Reports have shown that candidate DNA vaccines encoding the spike protein stimulated an immune response. This led to the study showing that injected mice are protected for SARS-CoV-1. Wang et al. have produced higher titres of neutralizing antibodies and demonstrated that major and minor neutralizing epitopes are located in the S1 and S2 subunits, respectively [32]. Other groups also found neutralizing epitopes in the S2 subunit [33,34]. Bisht et al. [35] have shown that intranasal or intramuscular inoculations of mice with highly attenuated Modified Vacciniavirus Ankara (MVA) vaccines encoding full-length SARS-CoV-1 S protein. This also produced neutralizing antibodies. Bukreyev et al. [36] reported that mucosal immunization of African green monkeys with an attenuated parainfluenza virus expressing S protein resulted in the production of neutralizing antibodies and protected animals from infection by challenge with SARS-CoV-1. These data suggest that the S protein can induce neutralizing antibodies and protective responses in immunized animals [37].Vaccines based on fragments containing neutralizing epitopes: fragments that were responsible for the virus binding to receptors within a host cell were targeted. Patients and animals that became infected with SAR-CoV-1 reacted strongly to this type of vaccine. They were immunized and inactivated with a receptor-binding domain (RBD) [29,38]. Absorption of antibodies by RBD showed the capability for removal of most of the neutralizing antibodies, RBD-specific antibodies isolated from these antisera have potent neutralizing activity [38,39]. The immunized mice were protected from SARS-CoV-1. The antibodies purified from the antisera against SARS-CoV-1 significantly inhibited RBD binding to ACE2 [29,38,40,41]. This suggested that RBD contains the major neutralizing epitopes in the S protein and is an ideal SARS vaccine candidate because RBD contains the receptor-binding site, which is critical for virus attachment to the target cell for infection [42,43,44]. Antibodies specific for RBD are expected to block the binding of the virus to the target cell. Therefore, RBD induces higher titers of neutralizing antibodies than those vaccines expressing the full-length S protein [31,32,35,37,43].

### 1.4. Vaccinations Proposed for MERS-CoV

In addition to the SAR-CoV-1 research that had taken place, the outbreak of the MERS-CoV allowed further research to be undertaken, below are some of the potential vaccination vehicles:Recombinant MERS-CoV: unlike the SARS-CoV-1 vaccine, the MERS-CoV vaccine was constructed based on the recombinant viruses using reverse genetics. This resulted in expressed marker mutations, which allowed for replication-competent, propagation-defective MERS-CoV vaccines [45].Viral-Vector-based MERS vaccine: this is similar to the vector-based SARS vaccine; MERS vaccines can also be constructed using viral vectors that express major MERS-CoV proteins, normally the S protein. Several such MERS vaccine candidates have been developed and/or tested for efficacy in mouse models or camels [46,47,48,49]. Viral vectors expressing full-length S protein of MERS-CoV induced S-specific antibody responses and/or T-cell responses in a mouse model via the intramuscular route, showed effective in vitro neutralization for MERS-CoV infection [47,50]. Additionally, vaccination of mice with an MVA-based full-length S vaccine-elicited MERS-CoV-specific CD8+ T cell response and neutralizing antibodies, protecting mice against MERS-CoV [48,49]. Intra-nasally or intra-muscularly administered MVA-S vaccine-induced mucosal immunity resulted in a significant reduction of excreted infectious viruses and viral RNA transcripts [45,46].Nanoparticle-based MERS vaccine: in recent years nanoparticles have been at the forefront of many research projects, this has allowed them to have the potential to develop a MERS vaccine. Nanoparticles containing MERS-CoV full-length S proteins can be prepared and purified from pellets of infected baculovirus insect cells. In the absence of adjuvants, these nanoparticles induced a lower level of MERS-CoV producing antibodies in mice. While in the presence of adjuvants, such as aluminium hydroxide (Alum) or Matrix M1, such antibodies were significantly increased and maintained. Thus, adjuvants are required for MERS nanoparticle vaccines and different adjuvants function differently in promoting the immunogenicity of these vaccines [45].DNA-based MERS vaccine: Like the full-length S gene of SARS-CoV-1, DNA encoding full-length S protein of MERS-CoV is utilized to develop MERS vaccines [31,51]. Indeed, intramuscular injections of mice with a synthetic DNA encoding full-length S proteins of MERS-CoV elicited potent virus-neutralizing antibodies and cellular immune responses, as represented by the secretion of INF-γ, TNF-α, and/or IL-2 cytokines in CD4+ and/or CD8+ T cells, as well as the production of antibodies in immunized camels.Subunit MERS vaccines: Protein-based subunit vaccines against MERS-CoV have been developed [52,53,54]. While some are designed on the basis of the full-length S1 proteins [55], the majority of them are based on viral RBD [53,54,56,57]. These RBD-based vaccines are evaluated for immunogenicity and protective immunity in a number of MERS-CoV mice models. The antigenicity and functionality of these RBD proteins have also been extensively investigated. In general, subunit vaccines might not induce immune responses as strong as those induced by other vaccine types mentioned above. However, the immunogenicity of subunit vaccines could be significantly promoted in the presence of an ideal adjuvant via an appropriate route [52]. In addition, it is also essential to maintain a suitable conformation of the protein antigen in the vaccine, such as the MERS-CoV RBD proteins [53,56]. In terms of safety consideration, subunit vaccines should be accounted as the safest vaccine type. They do not contain viral genetic materials, but only include essential antigens for eliciting protective immune responses, thus excluding the possibility of recovering virulence or inducing adverse reactions [58,59,60].

### 1.5. Vaccinations Proposed for SARS-CoV-2

The vaccination list below (Table 1) is of the candidates that have reached phase III clinical trials, these vaccinations use different technologies, the below table signifies the vaccinations that are in clinical trials.

## 2. Drug Research and Development: Promising Analogue to Patient

### 2.1. Drug Development and Discovery

The first step in the path to finding and identifying a compound that can be therapeutically useful, is the understanding of a disease or ailment. Once identified, the drug development and discovery process then includes identification of a suitable candidate, synthesis, analysis, validation and optimization for therapeutic effect before it can go to the next step. Once all these steps are accomplished and satisfied, the next step will be drug development. Drug discovery and development is a time consuming and financially expensive process, mainly due to the research and development that goes into identifying this one treatment. On average, it takes around 12–15 years to develop. The average cost is around $1 billion. This figure includes the high number of failures, usually in the hundreds if not thousands: for every successful drug molecule that has gone into market. The molecule chosen is identified from around 5000–10,000 compounds that did not pass through testing [67]. The process of drug development and discovery requires a high amount of resources from the best scientific minds, to highly equipped labs and technologies. It also takes some persistence and a slight piece of luck, however, the results, in the end, can be rewarding and can potentially lead to the beginning of a new medicine that can bring relief to billions of people [68].

### 2.2. Target Identification

Target identification requires the full analysis and understanding of the biological origins of a disease. This then requires the identification of potential targets, which is linked to identifying a therapeutic target and its role in tackling the disease [69]. An ideal target should show an improvement on current medicines and treatments in terms of efficacy, safety, clinical and financial feasibility to render it a viable alternative in terms of a new treatment to cost ratio, proving there is a major improvement to justify potentially higher costs. Target validation includes looking into Structure–Activity Relationships (SAR) of the new molecule, generating a drug, resistant to alterations within the target site and producing a toxic-free drug analogue capable to treat ailments [70]. Target validation is the process of demonstrating the functional role of the identified target in the disease phenotype.

Once the lead is identified, this compound should be stable, feasible, specific to the target site and have a high affinity and is selective to the target cell/organ. This is analysed through an in vivo test for efficacy and target engagement. This step includes the following [71]:SAR defined;Drug ability (preliminary toxicity);Synthetic feasibility;In-Vitro assessment of drug resistance and efflux potential;Evidence of in vivo efficacy of chemical class;Pharmacokinetics of chemical entity.

### 2.3. Lead Optimization

The next step in the drug discovery and development process is when the identification of the lead compound is complete. The optimization process requires the synthesis and characterization of the potential drug. This is in order to build a library of information for how this compound’s chemical structure and activity are related towards treating an ailment. This includes looking into interactions of targets and their metabolism. Selectivity and binding mechanisms are a key feature when deciding the identity of a promising lead, this would comprise towards the late stages of the drug development and discovery process. The main purpose of this step is to improve the properties of the lead target without affecting its structure that is effective in treating desired ailments. This is important, as specificity and selectivity are crucial factors in determining whether target lead compounds pass through the stages of the drug development process. Once the lead compound undergoes optimization and the pharmacodynamics, pharmacokinetics and toxicological analysis are undertaken, this would allow the researcher to find the optimum route of administration. To speed up the process high throughput drug metabolism studies are assessed, in addition to the previously mentioned studies. These help with characterizing the in-vivo behaviour of the lead compound [72].

### 2.4. Formulation and Development Process

Once a lead compound has been analysed, characterized and has shown preferable or improved treatment potential, it would mean a lead administration route has been identified and the formulation step would then allow for the preparation of a drug for the target route. This will allow for the production of a chemically stable, bioavailable and optimal dosage form. This process firstly consists of the pre-formulation step to test the chemical stability of a compound in different media, the dissolution potential of the active ingredient, the stability of the compound within these media and solid-state properties using Differential Scanning Calorimetry (DSC). The second step is the formulation component looking at routes to optimize the existing formulation, looking into new techniques to improve bioavailability; this may include the use of new delivery systems to show controlled or sustained release. Once a compound has shown the potential that it may be optimized, using the steps above the lead compound will be moved on to the pre-clinical stage of research.

### 2.5. Pre-Clinical Research

Pre-clinical testing is the movement of potential target lead into animal testing which will give the researcher an idea of how this lead may behave in humans. These tests are mainly to gauge the drug safety and efficacy. These preclinical tests have to demonstrate safety in order for regulatory authorities to give approval. These steps require high ethical standards as it is dealing with live animals. The approval is only granted to those processes that are undertaken correctly and show justification for clinical trials. There are two different types of tests that are assessed to achieve the aforementioned goals: (i) general Pharmacology tests and (ii) Toxicology tests. Pharmacology tests deal with pharmacodynamics and pharmacokinetics of drugs within animal subjects. This constitutes the analysis of unwanted pharmacological responses. This analysis shows the absorption, distribution, metabolism and elimination of the lead compound. Toxicological tests show the drug performance within animal subjects via in-vitro and in-vivo tests, these help to investigate the toxicity of the lead compounds on cells and organs. Depending on the target site and type of ailments, appropriate animal species are selected [73].

### 2.6. Clinical Research

Clinical research trials are the next step on the route for a drug to go onto the market. Once the pre-clinical tests are performed and the lead compound has been approved and shown to be safe and efficacious on animals, the clinical research phase allows the drug to then be moved on to human volunteers. Clinical studies follow a strict and specific protocol, as little mistakes can be life-altering to a human subject. A researcher must make it clear, what they intend to achieve from the clinical trials and what results are hypothesized through the different clinical phases. These include selecting what characteristics and criteria they would like as patients, number of people to take part, duration of the study, assessing the parameters and how data are going to be collated and analysed [74].

#### 2.6.1. Phase 0

The purpose of “Phase 0” tests is to verify the effect of the drug using micro-doses—which is performed through a single sub-therapeutic dose given to around 20 volunteers. This would give data to pharmacokinetics without the exertion of a potential pharmacological action. This analysis allows for the selection of a suitable drug that has no side effects at comparatively low doses. These stages take a couple of months, with around 90% of the analyzed drugs going to the next phase [75].

#### 2.6.2. Phase 1 (Safety and Dosage)

During phase 1 of the studies, 20 to 100 volunteers are selected. These can be either healthy volunteers, or ones with the condition. If however a compound is designed for cancer patients, then researchers will conduct the Phase 1 studies on patients with that type of cancer. These studies are closely monitored as tests are performed for drug interactions within the human body. Additionally, researchers adjust dosages according to the data obtained from the animal studies as this will allow for the analysis of tolerance levels—i.e., what a body can tolerate and what the acute side effects are. This phase gives direct answers on a small scale to the pharmacokinetics of the drug in the body, the side effects of increasing a dose and early information on how effective it is within human subjects. This study usually takes several months to complete. Around 80% of compounds go to the next phase [75].

#### 2.6.3. Phase 2 (Efficacy and Side Effects)

When the target compound passes Phase 1 testing, it is moved on to Phase 2. This phase tests for the efficacy and side effects of the compound, the testing is carried out on several hundred individuals with the target condition to be treated. Using a small batch of people, this phase shows the effective potential of the compound. Phase 2 studies indicate to researchers additional safety data they may not have attained from Phase 1. These data help researchers to develop more pertinent questions and methods that can be used for Phase 3. Phase 2 research may last for a couple of months and up to a couple of years, with almost 33% of drugs going through into Phase 3 [75].

#### 2.6.4. Phase 3 (Efficacy and Monitoring of Adverse Reactions)

Phase 3 testing assesses the efficacy and side effects on a larger amount of volunteers. Once Phase 2 is complete, successful compounds are given to around 300 to 3000 volunteers who have the disease or condition. This is a pivotal phase as it can now tell a researcher if the treatment has benefit to the target populous. Phase 3 provides the highest degree of safety data due to the high number of volunteers. This phase can show if there are less common side effects. Due to these studies averaging between 1 to 4 years, testing would more than likely show long term or rare side effects if there were to be any. Around 30% of compounds pass this phase of testing [75].

#### 2.6.5. Phase 4

Phase 4 is where the safety and efficacy tests are undertaken on several thousands of participants who carry the ailment intended for treatment. These are carried out post-FDA approval and during post-market monitoring [75].

## 3. FDA Review

Once a drug has gone through the aforementioned phases, the researcher may apply for a new drug application (NDA). This form will show and indicate all the information for the drug; its purpose is to show that the drug is safe and effective for the intended use. A researcher must include all the results obtained from pre-clinical to phase three trials. This information includes [75]:Proposed labeling;Safety updates;Drug abuse information;Patent information;Any data from studies that may have been conducted outside the United States;Institutional review board compliance information;Directions for use.

Once the FDA reviews the application, the review team will then look for the completeness of the application—which will give way to the final acceptance or refusal of the application. If the NDA is complete, it may take up to 10 months to accept the application. The process of analysis includes the following [75]:Each member of the board conducts a full review of their section of the application.FDA instructors travel to clinical study sites to conduct an inspection of the facilities. The FDA looks for evidence of fabrication, manipulation or withholding of data.The project manager assigned will oversee all the individual reviews into a combined action package. The review team recommends a decision and a senior figure will make the final decision.

## 4. FDA Approval

Once the FDA reviews and accepts an application, it is necessary for the FDA to work with the applicant to develop and refine prescribing information. Often, any remaining issues are resolved prior to the drug being approved for market release. Sometimes FDA requires the developer to address questions based on existing data. In other cases, FDA requires additional studies. At this point, the developer can decide whether to continue further development or not. If a developer disagrees with an FDA decision, then there are mechanisms for formal appeal [75].

## 5. FDA Post Market Drug Safety Monitoring

Even with all the information obtained to this point through the research process and clinical trials of a lead compound, the complete picture of a drug can sometimes take months and even years after the compound is released into the market. The FDA monitors these newly released treatments in the marketplace and reviews reports with any problems encountered. This can help to identify and eliminate any additional problems via the information accrued about the dosage form and its usage [75].

## 6. Conclusions

This review took a detailed look into each step of the drug development process. It shows that the average cycle for “Research into Lead compounds > Approval by the FDA > Drug going into market” takes around 17 to 20 years. Although the COVID-19 vaccination took a fraction of that time, the reasons for this may be justified.

COVID-19 belongs to a familiar viral family that has been around for decades. This means much of the above-ground has been inherently covered, by which many scientists have been able to acquire existing data belonging to that family of viral infections—hence they have been able to bring out the vaccine projects that were worked on previously. Many previous projects researched into the original SARS virus, which was present in 2003. These projects came to a halt as the previous virus killed only 800 people and infected a small population. Earlier research showed that the spike proteins exhibited particular interest for targeting, via a vaccine, therefore giving scientists a foundation to work on.

Their reason for stopping the research of the SARS-CoV-1 virus is that these infections caused were of the acute variety. This means that—for most cases—they will be asymptomatic and their immune system will be able to fight off the virus, therefore recovering without the requirement of a suitable treatment. Other vaccinations were therefore given higher priority (for research) such as a Human Immuno-deficiency Virus (HIV) vaccine, as HIV causes a chronic infection; hence, a natural immune response is not enough to kill that virus. Henceforth, the data that collated for the potential vaccine remained on record. This information obtained was eventually used for the vaccination that was formulated for SAR-CoV-2 as was shown in this review.

The COVID-19 vaccine came in double quick time due to the use of some innovative biotechnological methods. This includes the formulation of a vaccine that is capable to adapt to different platforms and its ability to maneuver from pathogen to pathogen if the virus was able to mutate and travel within the body. Older techniques simply targeted the virus in order to weaken it, so as to inactivate the target virus. The new biotechnology methods can be used due to the previous information scientists were able to collate for the DNA of this viral family. Pfizer, in collaboration with BioNTech, was able to utilize their inherent knowledge and advance that research at a rapid pace, identifying an analogue quickly and fast-tracking it through the relevant stages of research.

Billions of dollars were/are still being invested into finding a vaccine and fast-tracking it at this rate. A reason for the slow research and development until a drug goes through FDA approval and onto the market is due to research companies sometimes being reluctant to invest large chunks of financial capital. Companies would like to see the lead compound go through each stage and satisfy each step before committing to further funding that research. The billions of dollars invested was not for just research, but also for companies to manufacture these whilst research was being undertaken, as companies were taking the high risk, high investment, high reward approach towards finding a vaccine for COVID-19. What this meant is that if a compound was found to be effective, potentially millions of vaccines can be distributed the next day, instead of waiting months to manufacture the large quantities required.

Finally, regulatory decisions are being decided quickly due to the desperate nature of the situation, which helps in advancing clinical trials at a rapid rate. Nevertheless, this may come at a cost. The speed of vaccine development can have adverse effects, resulting in long-term studies and years of post-vaccine treatment. However, as is the case with most treatments, the benefits outweigh the risks and far more people—and to greater extent countries—need the vaccine, as economies have taken large hits, with livelihoods being crippled. With families losing a large number of relatives due to the virus, this vaccination is capable of restoring quality of life to hundreds of millions if not billions of people worldwide. The vaccination may have come more rapidly than normal, but could well be justified due to the reasons explained in this review. This vaccine will have a long-term effect on how research is viewed and can be undertaken for new vaccinations and other treatments. New strategies have been learned and have enhanced the effectiveness of processes within the research models for future therapies, finally speeding up potential treatments or therapies from being an interesting agent to being administered to the patient.

## Figures and Tables

**Figure 1 vaccines-09-00120-f001:**
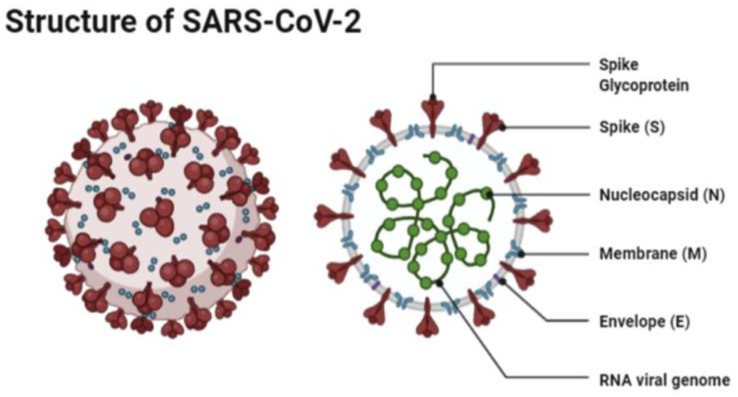
The structure of SAR-CoV-2 under a microscope as illustrated by Agarwal et al. [18].

**Figure 2 vaccines-09-00120-f002:**
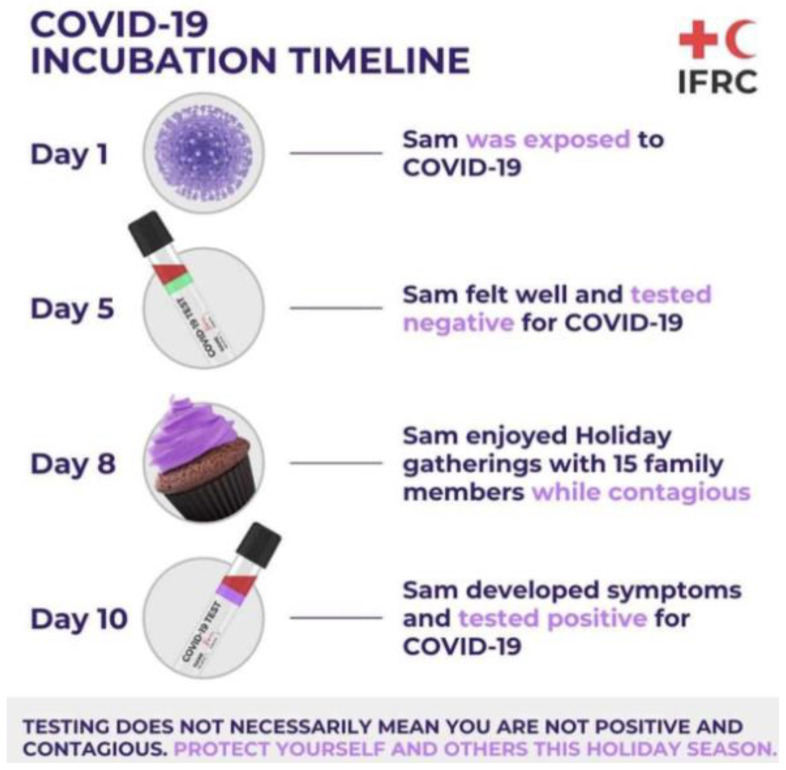
Illustration of a potential problem that is faced with RT-PCR testing due to difficulties in early detection of SAR-CoV-2 as described by the Red Cross [25].

**Figure 3 vaccines-09-00120-f003:**
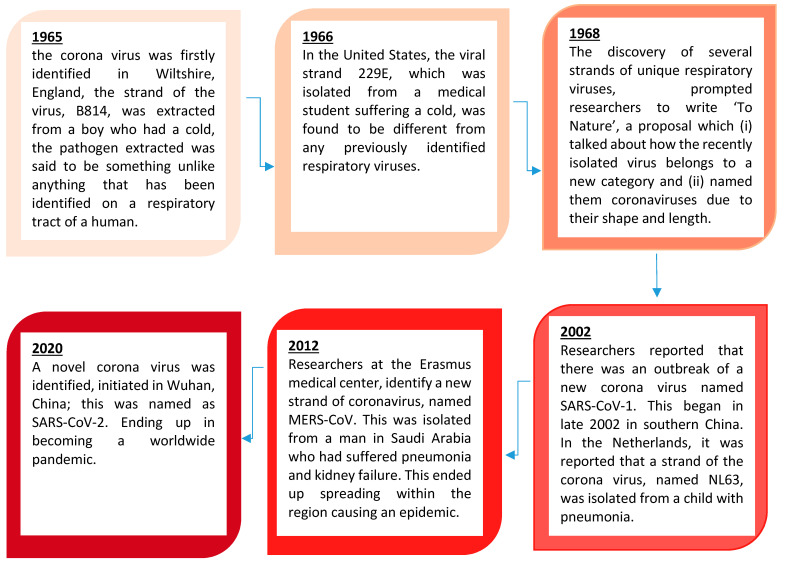
Brief timeline history of the Ortho-Coronavirinae as illustrated by Williams 2020 [26].

**Table 1 vaccines-09-00120-t001:** Vaccinations currently in phase III of clinical trials for the SARS-CoV-2 virus.

Vaccine Candidate, Developers	Technology Used	Current Phase	Completed Phase (Findings)	Clinical Trial Sites
AZD1222 University of Oxford, AstraZeneca [61]	Modified chimp adenovirus vector (ChAdOx1)	Phase III (30,000) Interventional; randomized, placebo-controlled study for efficacy, safety, and immunogenicity. Brazil (5000) International enrolment of the Phase III trial was paused on 8 September 2020, due to an adverse neurological event in one participant, but resumed on 12 September in the UK. On 23 October, AstraZeneca said it will resume the trial in the US	Phase I-II (543) Spike-specific antibodies at day 28; neutralizing antibodies after a booster dose at day 56. Adverse effects: pain at the injection site, headache, fever, chills, muscle ache, malaise in more than 60% of participants; paracetamol allowed for some participants to increase tolerability	20 in the UK, São Paulo
Ad5-nCoV CanSinoBIO, Beijing Institute of Biotechnology of the Academy of Military Medical Sciences [62]	Recombinant adenovirus type 5 vector	Phase III (40,000) global multi-center, randomized, double-blind, placebo-controlled to evaluate efficacy, safety and immunogenicity in Mexico, Pakistan, Russia, Saudi Arabia	Phase II (508) Neutralizing antibody and T cell responses. Adverse effects: moderate over 7 days: 74% had fever, pain, fatigue	China and Pakistan
BNT162b2 BioNTech, Fosun Pharma, Pfizer [63]	mRNA	Phase III (30,000) Randomized, placebo-controlled	Phase I-II (45) Strong RBD-binding IgG and neutralizing antibody response peaked 7 days after a booster dose, robust CD4+ and CD8+ T cell responses, undetermined durability. Adverse effects: dose-dependent and moderate including pain at the injection site, fatigue, headache, chills, muscle and joint pain, fever	62 in the USA and Germany
CoronaVac Sinovac [64]	Inactivated SARS-CoV-2	Phase III (33,620) Double-blind, randomized, placebo-controlled to evaluate efficacy and safety in Brazil (15,000); Chile (3000); Indonesia (1620); Turkey (13,000) Brazil paused Phase III trials on November 10 after the suicide of a volunteer in the trials before resuming them on November 11.	Phase II (600) Preprint. Immunogenicity eliciting 92% seroconversion at lower dose; Adverse effects: mild in severity, pain at injection site	2 in China; 22 in Brazil; Bandung, Indonesia
mRNA-1273 Moderna, NIAID, BARDA [65]	Lipid nanoparticle dispersion containing mRNA	Phase III (30,000) Interventional; randomized, placebo-controlled study for efficacy, safety, and immunogenicity	Phase I (45) Dose-dependent neutralizing antibody response on two-dose schedule; undetermined durability. Adverse effects: fever, fatigue, headache, muscle ache, and pain at the injection site	89 sites in the USA
Ad26.COV2.S Janssen Pharmaceuticals (Johnson and Johnson), BIDMC [66]	Non-replicating viral vector	Phase III (60,000) Randomized, double-blinded, placebo-controlled Temporarily paused on 13 October 2020, due to an unexplained illness in a participant. Johnson and Johnson announced, on 23 October, that they are preparing to resume the trial in the US.	Phase I-II (1045) Preprint. Seroconversion for S antibodies over 95%. Adverse effects: injection site pain, fatigue, headache and myalgia	291 in US, Argentina, Brazil, Chile, Colombia, Mexico, Peru, Philippines, South Africa and Ukraine

## Data Availability

Not applicable.

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
