# Peer review of "COVID-19 Vaccination: From Interesting Agent to the Patient"

_vaccines, 2021, doi:10.3390/vaccines9020120_

Round 1
Reviewer 1 Report
Dear authors,
Many thanks for this interesting article. However, I would like to raise a few points.
Minor: Line numbering would have faciliated the review.
Major:
Although being a review to a lot of references are missing for claims being made.
Abstract:
line 3: Why woult it ease 60 million infected people. The infected usually are contagious for 10 days and a vaccine will not be able to lower their morbidity nor mortality.
line 4: Why do you use we when there is only one author.
Introduction:
line 2 "the Coronavirus" this is language of people not being familiar with medicine.
"scenario being in schools where those suffering symptoms" Is there any evidence for this. In most countries transmission is in the private sector (family, friends etc.). Is there a reference?
Further sections:
Summarizing of the main findings in Tables would be beneficial.
Discussion:
The discussion is too short in my opinion.
"Billions of dollars have been thrown into finding a vaccine" seems to have a very negative connotation.
Author Response
Dear Reviewer
Happy holidays, thanks for your remarks, i have addressed the comments that you have highlighted.
Firstly, apologies for not numbering the lines, this should have been done prior to submission, i have done so now, hopefully this will be easier for you to look at and critique.
Secondly, i have added some references to some of the claims that were being made. I have also adjusted some of the wording, this review is meant for non-scientists in addition to scientists, so therefore the language used was basic english, so that all pupils from all professions can read and understand the material.
I have also tabulated the latest findings for the SARS-CoV-2 vaccinations that are being researched or are into mass production.
The discussion and conclusion part sounded more like a conclusion only, so, to satisfy this point i have renamed the section just as a conclusion as this is a review giving concluding remarks on the latest vaccination and why it may be a suitable vaccination.
I have removed billions of dollars and replaced it with have been invested, to make it sound less negative.
I hope that your points have been addressed, and I thank you again for giving your time and effort in reading and providing feedback on the review.
Please find attached the amended version for the review article.
Happy new year!
Best Regards
Dr. Anis Daou
Reviewer 2 Report
The manuscript by Daou provided a detailed review of the novel Coronavirus (COVID-19); however, it remains unorganized in many of its sections. The title is unclear and requires authors to clearly focus on a better title that identifies the main contribution of this review, particularly if too much information described the introductory details of COVID-19.
Abstract:
- Line 4-5: The number of affected people should be updated in accordance with the new statistics available. The number of affected people is more than 73.5 million.
- Line 8: The author should make clear if the human respiratory organs are the only affected organs or there are other organs that may be affected by SARS-CoV-2 (#check doi: 10.1016/j.biopha.2020.110195, do: 10.1007/s10238-020-00648-x, and do: 10.1084/jem.20050828).
- Line 11-12: It remains hard to draw this conclusion, particularly if all available vaccines are still under clinical trials and benefit from using previously available toolkits of vaccination to other viruses.
.............................................................................................................
- The introduction should be concise and describe clearly the objectives of this review.
- P-2, Line 6: RT-PCR is used for the diagnosis of SARS-CoV-2 and not for genomic analysis. Genomic analysis requires a very sensitive sequencing approach to differentiate the virus and identify different virus mutants if any.
- P-2, Line 12: The author should describe in detail the unique replication machinery of SARS-CoV-2, particularly if it is most relevant to the review topic.
- P-3, Line 2-5: I think several other characteristic pathologies were recognized in the symptomatic state that authors should add (#check my previous comments related to this same point in the abstract section).
- Figure 1: The author should provide a detailed description of the virus structure in the figure caption.
- P-6, Line 6, and line 11: add the correct citation herein.
- P-6: I think the author should provide some information about the problems associated with the diagnosis of SARS-CoV-2. This area still has gaps in knowledge that require further details.
- Figure 2: I suggest adding more effort to this figure to make it more attractive and detailed. For example, the dates should be bold, and more focused information should be included.
- Authors should add examples of the drugs used for the treatment early and how these regimens evolved over time for better uses and application.
- The latter point should also be discussed for vaccination. The author should describe a list of all available candidate vaccines and provide some evidence for the most important candidates from recently published studies.
- The drugs and vaccination details are the most crucial points that should be detailed in this review to give feelings that the author reviews properly these points and not just provide a collection of data from diverse sources. I would love to see a more concise review with more detailed information available in the most updated publications.
- In summary, This manuscript should clearly define the available toolkits and identify the gaps in knowledge of treatment and vaccination of SARS-COV2. These points are clearly missed across this review. So, I would love to see more details and a better presentation to this review. There are many areas for improvements that I suggested in my comments above, particularly improvements of figures, including tables that list the candidate drugs and vaccines available in diverse phases or even in the market.
Author Response
Dear Reviewer
Happy holidays, thanks for your remarks, i have addressed the comments that you have highlighted.
Firstly, i have renamed the review article to a more focused title, incorporating the corona virus aspect into it.
Secondly, i have addressed all the points made, adding points in the abstract, adjusting the wording to better fit the amount of number of pupils who have been effected by the virus, in addition i have added other symptoms that may be caused by the virus, and re-adjusted the final sentence.
Thirdly, the introduction section, i have taken into account all the points you have suggested and adjusted the accordingly, I have added a table with all the current research for the vaccination for covid-19, and the difficulties in identifying the virus through RT-PCR, as you have suggested. i have also made the figure more attractive by adding colour and adjusting font. i have also given a more detailed description of the structure of the virus and added some other symptoms that may be felt by sufferers.
I have also adjusted some of the wording, this review is meant for non-scientists in addition to scientists, so therefore the language used was basic english, so that all pupils from all professions can read and understand the material.
I hope that your points have been addressed, and I thank you again for giving your time and effort in reading and providing feedback on the review.
Please find attached the amended version for the review article.
Happy new year!
Best Regards
Dr. Anis Daou
Round 2
Reviewer 1 Report
Dear authors,
many thanks for your revised version of the manuscript.
However, I would like to politely ask if there is a permission for the incorporation of the figures?
Best regards
Author Response
Happy new year.
Hope you are doing well, the images are on journal open to the public, in addition they are fully referenced just in case a reader wants to refer to those articles.
Best Regards

Reviewer 2 Report
- The style of references inside the text should be changes into the Journal style. For example, references should be at the end of each sentence and not at the beginning of the next sentence. Please look carefully to the Journal style.
- The manuscript should be considered for further copy editing of English language; there are many linguistic problems across the manuscript text.
Author Response
Hi
Happy new year. Thanks for your comments, I have adjusted the references according to journal protocol. In regards to your second comment about language, I did forward it to an English editing service, but requested that the language used isn't too advanced so that it can reach, and be understood, by a larger audience. they did fix some punctuation and some structures of sentences. Hope this answers your queries.
Best Regards
Dr. Anis Daou PhD
